# PLA2R-IgG4 antibody as a predictive biomarker of treatment effectiveness and prognostic evaluation in patients with idiopathic membranous nephropathy: a retrospective study

Yiqing Huang[1,*], Junyi Zhou[1,*], Kezhi Zhou[1], Biao Huang[2], Jing Xue[1], Xiran Zhang[1], Bin Liu[1], Zhijian Zhang[1], Leting Zhou[1], Ting Cai[1], Yi Zhang[3], Zhigang Hu[4], Liang Wang[1] and Xiaobin Liu[1]

[1] Department of Nephrology, The Affiliated Wuxi People's Hospital of Nanjing Medical University, Wuxi, China
[2] College of Life Sciences and Medicine, Zhejiang Sci-Tech University, Hangzhou, China
[3] NHC Key Laboratory of Nuclear Medicine, Jiangsu Key Laboratory of Molecular Nuclear Medicine, Jiangsu Institute of Nuclear Medicine, Wuxi, China
[4] Medical Laboratory, The Affiliated Wuxi Children's Hospital of Nanjing Medical University, Wuxi, China
* These authors contributed equally to this work.

Corresponding authors
Liang Wang, wlwxsnk@163.com
Xiaobin Liu, lxbwxsnk@163.com

## ABSTRACT

**Background**. The Kidney Disease Improving Global Outcomes (KDIGO) 2021 guidelines recommend Rituximab (RTX) as the first-line therapy and phospholipase A2 receptor (PLA2R) antibody as a biomarker for remission and prognosis in patients with idiopathic membranous nephropathy (IMN).

**Methods**. This study was a retrospective analysis of 70 patients with IMN treated with either rituximab (RTX) or cyclophosphamide (CTX) and steroid. Quantitative detection of PLA2R-IgG and PLA2R-IgG4 antibodies at sixth month after treatment, determined using time-resolved fluoroimmunoassay (TRFIA), were used for treatment effectiveness analysis and prognostic evaluation in patients with IMN.

**Results**. After 12 months of therapy, the remission rate of proteinuria, including complete remission (CR) and partial remission (PR) in the RTX group and the CTX group, were 74% versus 67.5% ($P = 0.114$), respectively. Both PLA2R-IgG and PLA2R-IgG4 levels were decreased in patients with remission of proteinuria after 6 months of therapy. Receiver operating characteristic curve (ROC) curve analysis exhibited that the AUC of PLA2R-IgG4 and the PLA2R-IgG as laboratory criteria for proteinuria remission were 0.970 versus 0.886 ($P = 0.0516$), respectively, after 6 months of treatment. The cut-off value of PLA2R-IgG4 was 7.67 RU/mL and the sensitivity and specificity of remission rate at 6th month were 90.9% and 100%, respectively. Furthermore, the AUC of the PLA2R-IgG4 and PLA2R-IgG to predict the outcome after 12 months of treatment were 0.922 versus 0.897 ($P = 0.3270$), respectively. With the cut-off value of PLA2R-IgG4 being 22.985 RU/mL, the sensitivity and specificity of remission rate at 12th month were 100% and 87.1%, respectively. Logistic regression analysis revealed that the PLA2R-IgG4 level ($P = 0.023$), the rate of decrease of PLA2R-IgG4 level ($P = 0.034$), and eGFR level ($P = 0.012$) were significantly associated with remission.

**Conclusions**. We found that the patients in the RTX group and CTX group achieved effective remission of proteinuria after 12 months of treatment. PLA2R-IgG4 may be a more effective biomarker for treatment effectiveness analysis and prognostic assessment, compared with anti-PLA2R-IgG for PLA2R associated IMN.

## INTRODUCTION

In 2009, *Beck Jr et al. (2009)* first identified circulating autoantibodies against phospholipase A2 receptor (PLA2R) in 70% of patients with idiopathic membranous nephropathy (IMN). Furthermore, IgG4 was found as the main subclass of autoantibodies in serum samples, which were bound to PLA2R in glomerular podocytes (*Beck Jr et al., 2009*). Kidney Disease Improving Global Outcomes (KDIGO) 2021 guidelines recommend anti-PLA2R antibody as a laboratory index for diagnosis, treatment effectiveness analysis, and prognostic evaluation (*Du et al., 2014*; *Hofstra et al., 2012*). A higher level of PLA2R antibody is more likely to have a worse renal function and therapeutic response (*Hoxha et al., 2014a*; *Hoxha et al., 2014b*). With the serum anti- PLA2R antibody becoming negative, it takes a long time to return the glomerular basement membrane (GBM) architecture to normal. Therefore, the PLA2R antibody level precedes the clinical remission and predicts clinical response in the course of spontaneous or treatment-induced remission (*Dahan et al., 2017*; *Jatem-Escalante et al., 2021*). According to KDIGO guidelines, longitudinal monitoring of PLA2R antibody levels during treatment can guide therapeutic adjustments (*Medrano et al., 2015*; *Wei et al., 2016*).

Until recently, the enzyme-linked immunosorbent assay (ELISA) and indirect immunofluorescence assay (IIFA) were the most widely used immunological methods for the detection of blood PLA2R antibodies. However, IIFA is a semiquantitative detection method, and ELISA has low sensitivity in detecting PLA2R antibody titer (*Behnert et al., 2014*; *Bobart et al., 2019*). In 2017, *Huang et al. (2017)* developed a more sensitive quantitative detection method, named time-resolved fluoroimmunoassay (TRFIA), which improved the sensitivity to 74% and specificity was 100% in diagnosing IMN. TRFIA is a novel non-radioactive microanalytical technique based on Eu3+ and their chelators with unique fluorescence properties as tracers. The addition of a fluorescence intensifier enhanced the original fluorescence by 1 million times and improved the sensitivity and the range of detection during TRFIA for measuring PLA2R-IgG (*Huang et al., 2017*). It is widely used in the clinical diagnosis and treatment of tumors, infectious diseases, endocrine diseases, autoimmune diseases, hereditary diseases. *Huang et al. (2019)* made a further improvement on the sensitivity to 90% by investigating PLA2R-IgG4 titer by TRFIA. *Liu et al. (2021)* demonstrated that PLA2R-IgG4 was a more efficient biomarker in predicting the risk of progression in IMN, compared with anti-PLA2R-IgG.

Presently, the typical treatment in patients with IMN who are at high risk of progressive kidney failure is prednisone combined with cyclophosphamide (CTX), which brings some serious side effects related to glucocorticoid or cytotoxicity related to cyclophosphamide (*Hofstra & Wetzels, 2010*; *Ponticelli et al., 1995*). Calcineurin inhibitors such as cyclosporine and tacrolimus have been proven to be effective alternative therapies in recent years. However, one of the most concerning characteristics of calcineurin inhibitors was the considerable relapse of a disease following discontinuation of therapy (*Qiu et al., 2017*). Rituximab (RTX) that showed similar efficacy and fewer side effects in a significant number of observational and randomized controlled studies, was considered as a novel treatment option for IMN (*Fernández-Juárez et al., 2021*; *Fervenza et al., 2019*; *Scolari et al., 2021*). Before the draft of the 2020 new guidelines, the diagnosis and treatment were according to the 2012 KDIGO guidelines. Immunosuppressive therapy is recommended for intermediate-risk patients who have not responded for 6 months expectant treatment or high-risk patients. In the selection of immunosuppressive agents, CTX is the first choice. Since the update of the new draft guideline of KDIGO in 2020, RTX is recommended as the first-line drug in intermediate-risk or high-risk MN patients, while CTX should be used preferentially in high-risk or extremely high-risk MN patients.

In this study, we aimed to retrospectively analyze the effect of either rituximab or cyclophosphamide and steroid in the treatment of PLA2R-associated IMN. Also, we planned to use TRFIA in identifying PLA2R-IgG and PLA2R-IgG4 antibodies in the 6th month after treatment and evaluate their use as sensitive laboratory biomarkers for remission and prognostic prediction following a 12-month treatment.

## METHODS

### Patients and blood samples collection

Patients who were diagnosed with IMN confirmed for PLA2R and mainly IgG4 subclass deposition in glomerular podocytes by renal biopsy and immunostaining from January 2016 to December 2020 in the Affiliated Wuxi People's Hospital of Nanjing Medical University were selected for the analyses. This study had been approved by the ethics committee of the Wuxi People's Hospital (ethical approval no. kyl2016001). Patients were also in moderate or high risk and unlikely to develop spontaneous remission. Patients who were positive for hepatitis B antigen, hepatitis C virus antibodies, malignancy, infections, and/or exposure to toxic agents that could induce MN were excluded from the analysis. Blood samples that were collected before the immunosuppressive therapy and after 6 months of treatment and stored at −80 °C after centrifuging at 3,000 rpm/min for 4 min were used.

### Treatment

The therapeutic regimen was decided by nephrologists after evaluating the risk of progressive loss of kidney function. All patients provided written informed consent. Thirty patients received a standard dose of RTX and forty subjects received CTX treatment. Initially, patients in the RTX group received either 1,000 mg intravenous infusions of RTX repeated once after two weeks or 375 mg/m$^2$/week for four weeks. Subsequently, patients

received RTX at the time of relapse or at 6 monthly intervals as a remission maintenance therapy. Patients in the CTX group received CTX intravenously with a dose of $0.6-0.8$ g/m$^2$ (maximum of 1g) monthly for six months and then once every 2 or 3 months combined with oral administration of prednisone.

### Criteria for proteinuria remission

Complete response (CR) was considered when proteinuria was $\leq$0.5 g/day with a normal or stable glomerular filtration rate (eGFR was evaluated using the MDRD formula). Partial response (PR) was considered when 24-h urine protein was <3.5 g/day with a reduction of at least 50%, albuminemia was >3.5 g/dL, and there was a stable renal function. No response was considered when a <50% reduction of proteinuria or worsening of serum creatinine was noticed.

### Quantitative detection of PLA2R-IgG and -IgG4

First, serum samples containing PLA2R-IgG and anti-PLA2R -IgG4 were diluted by 1: 200 and 1: 20, respectively. Then, 100 $\mu$L of standard or diluted serum samples were pipetted to the 96-well microtiter plates prepared already. And the plates were placed on the incubator shaker at 25 °C for 1 h. After blocking non-specific substrates, europium-labelled goat anti-human IgG or mouse anti-human IgG4 antibodies were added to the plates and then shaken at 25 °C for 1 h. After rinsing for 6 times, 200 $\mu$L enhancement solution was added to the 96-well plates and the plates were agitated for 5 min. Finally, AutoDELFIA1235 plate processor and the MultiCalc software application were used to determine and analyze the concentrations of serum samples of anti-PLA2R-IgG and -IgG4 to analyze. The more specific details were provided in previous studies (*Huang et al., 2017*; *Huang et al., 2019*).

### Statistical analysis

Means and standard deviations together with median and quartiles were used for descriptive statistical analyses. For continuous variables, the $t$-test or Mann–Whitney $U$ test was used for assessing any difference between the two groups, while the Chi-square test was used for dichotomized variables. A $p$-value of <0.05 was considered statistically significant. The Spearman rank-order correlation combined with logistic regression analysis were used to identify the variables associated with the outcome measures. To determine the sensitivity and specificity of PLA2R-IgG and PLA2R-IgG4 titers detected by TRFIA after 6 months of therapy as laboratory criteria for remission and the prediction of outcome after 12 months of treatment, receiver operating characteristic (ROC) curves were drawn and the areas under the ROC curves (AUCs) were calculated. Statistical analysis was performed using SPSS version 26.0.

## RESULTS

### The characteristics of patients in the RTX group and the CTX group at baseline

The baseline characteristics of the patients were similar in the two groups (Table 1). Glomerular staining of PLA2R protein were all positive in the two groups of patients. What's more, the mainly IgG subclass were IgG4. However, patients in the RTX group

**Table 1  The characteristics of patients at baseline.**

| Characteristics | Rituximab ($n = 30$) | Cyclophosphamide ($n = 40$) | t, X$^2$, or Z-value | p-value |
|---|---|---|---|---|
| Male sex no. (%) | 17 (56.7) | 23 (57.5) | 0.005 | 0.944 |
| Age, year | 54.23 ± 12.16 | 50.63 ± 11.77 | 1.121 | 0.266 |
| Hypertension no. (%) | 19 (63) | 27 (68) | 0.132 | 0.716 |
| Diabetes no. (%) | 7 (23) | 7 (18) | 0.365 | 0.546 |
| Systolic BP, mmHg | 127.03 ± 17.19 | 133.00 ± 18.65 | −1.369 | 0.175 |
| Diastolic BP, mmHg | 80.40 ± 9.89 | 83.23 ± 11.94 | −1.053 | 0.296 |
| ACEI/ARB use no. (%) | 25 (83) | 37 (93) | 1.423 | 0.233 |
| Tubulointerstitial lesion score[a] | 2.90 ± 1.32 | 3.08 ± 1.26 | −0.517 | 0.607 |
| Pathological stage[b] | 1.50 ± 0.509 | 1.58 ± 0.594 | −0.555 | 0.581 |
| Glomerular PLA2R-IgG4 score[c] | 2.93 ± 0.25 | 2.85 ± 0.36 | 1.078 | 0.285 |
| Serum cholesterol, mmol/L | 6.95 ± 1.85 | 7.85 ± 1.09 | −1.887 | 0.063 |
| Serum triglycerides, mmol/L | 2.18 ± 1.69 | 3.08 ± 1.83 | −2.108 | 0.039 |
| Uric acid, mmol/L | 363.86 ± 82.92 | 360.24 ± 83.83 | 0.192 | 0.848 |
| Serum creatinine, umol/L | 91.144 ± 2.61 | 77.16 ± 20.06 | 1.665 | 0.104 |
| eGFR, mL/min per 1.73 m$^2$ | 81.44 ± 28.05 | 92.98 ± 18.09 | −1.966 | 0.055 |
| Serum albumin, g/L | 23.46 ± 6.12 | 21.30 ± 4.85 | 1.648 | 0.104 |
| Proteinuria, g/L | 4.10 ± 1.67 | 5.71 ± 2.01 | −3.566 | 0.001 |
| PLA2R-IgG, RU/mL[d] | 71.49 (9.19, 225.88) | 55.04 (17.62, 86.78) | −0.668 | 0.504 |
| PLA2R-IgG4, RU/mL[d] | 508.33 (58.15, 1874.84) | 398.57 (81.55, 824.34) | −1.059 | 0.289 |

**Notes.**

Values are expressed as means ± SD or n (%).

[a] Tubulointerstitial lesion score was a sum of renal tubular atrophy score, renal interstitial fibrosis score, and renal interstitial lymphoplasmacytic infiltrate score.

[b] Pathological stage classified according to Enrenreich and Churg standards.

[c] Glomerular PLA2R-IgG4 score was immunofluorescence intensity(3=+++,2=++,1=+,0=-).

[d] PLA2R-IgG and PLA2R-IgG4 were detected by TRIFA and expressed as the interquartile range (IQR).

had higher serum creatinine and relatively lower eGFR compared with the CTX group although no significant statistical differences were found ($P > 0.05$). Interestingly, the CTX group had a higher 24-h urine protein level than that of the RTX group (4.10 ± 1.67 versus 5.71 ± 2.01 g/24-h; $P = 0.001$), while a significant statistical difference in serum albumin between the two groups was not found (13.46 ± 6.12 versus 21.30 ± 4.85 g/L; $P = 0.104$). Since the PLA2R-IgG and PLA2R-IgG4 titer distribution curves did not obey a normal distribution the Mann–Whitney $U$ test was used to analyze, and it showed no significant difference between the two groups. Three patients did not adhere to the follow-up protocol during the 12-month study period.

### Effectiveness analysis of the patients in the two groups

The per-protocol analysis showed that there were no statistical differences in the remission rate of proteinuria, including CR and PR, between the RTX group and the CTX group at 6th month (44.8% versus 55%, $P = 0.497$), 9th month (66.6% versus 65%, $P = 0.278$) and 12th month (74% versus 67.5%, $P = 0.114$) (Table 2). The 24-h urine protein was decreased significantly both in the CTX group and the RTX group at 6th month (2.53 ± 1.73 versus

**Table 2 The remission rate of proteinuria in the two groups.**

| Time | Rituximab | | Cyclophosphamide | | X²-value | P-value |
|---|---|---|---|---|---|---|
| | CR (%) | PR (%) | CR (%) | PR (%) | | |
| 6th month | 4/29 (13.8) | 9/29 (31.0) | 10/40 (25) | 12/40 (30) | 1.400 | 0.497 |
| 9th month | 6/27 (22.2) | 12/27 (44.4) | 15/40 (37.5) | 11/40 (27.5) | 2.562 | 0.278 |
| 12th month | 10/27 (37.0) | 10/27 (37.0) | 21/40 (52.5) | 6/40 (15.0) | 4.344 | 0.114 |

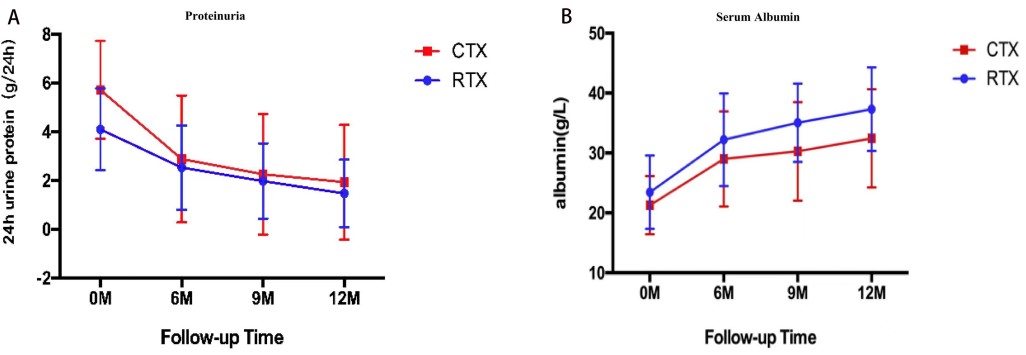

**Figure 1 Temporal changes in proteinuria (A) and serum albumin (B) in the RTX group and the CTX group (Means ± SD).**

2.88 ± 2.6 g/24-h, $P = 0.499$), and 12th month (1.47 ± 1.38 versus 1.92 ± 2.35 g/24 h, $P = 0.322$), which was higher in the CTX group than that of the RTX group at baseline (Fig. 1A). The serum albumin manifested a significant increase in both groups at 6th month (32.22 ± 7.73 versus 2.88 ± 2.6 g/L, $P = 0.100$), and 12th month (37.30 ± 6.96 versus 32.45 ± 8.19 g/L, $P = 0.014$), which was similar in the two groups at baseline (Fig. 1B). A significant decrease of PLA2R-IgG level was observed at month 6 (4.67 RU/mL, (IQR, 1.82–15.14), $P = 0.000$), compared with baseline (71.49 RU/mL (IQR, 9.19–225.88), $P = 0.000$) in the RTX group and at month 6 (4.83 RU/mL (IQR, 2.34–82.49), $P = 0.000$), compared with baseline (55.04 RU/mL (IQR, 17.62–86.78)) in the CTX group (Fig. 2A). A clear reduction in PLA2R-IgG4 level at month 6 (4.83 RU/mL (IQR, 2.34–82.49), $P = 0.000$), compared with baseline (508.33 RU/mL (IQR, 58.14–1874.83)) in the RTX group and at month 6 (5.78 RU/mL (IQR, 2.16–176.24), $P = 0.000$), compared with baseline (398.57 RU/mL (IQR, 81.55–824.34)) in the CTX group was found (Fig. 2B).

Patients who achieved CR and PR of proteinuria at 6th month showed a lower PLA2R-IgG and PLA2R-IgG4 levels compared with those who had no response to the immunotherapy (4.44 RU/mL (IQR, 1.95−7.11) versus 16.78 RU/mL (IQR, 11.61–54.17) of PLA2R-IgG, $P = 0.000$, and 2.63 RU/mL (IQR, 0.93−3.82) versus 144.48 RU/mL (IQR, 34.98–305.61) of PLA2R-IgG4, $P = 0.000$) (Figs. 2C, 2D).

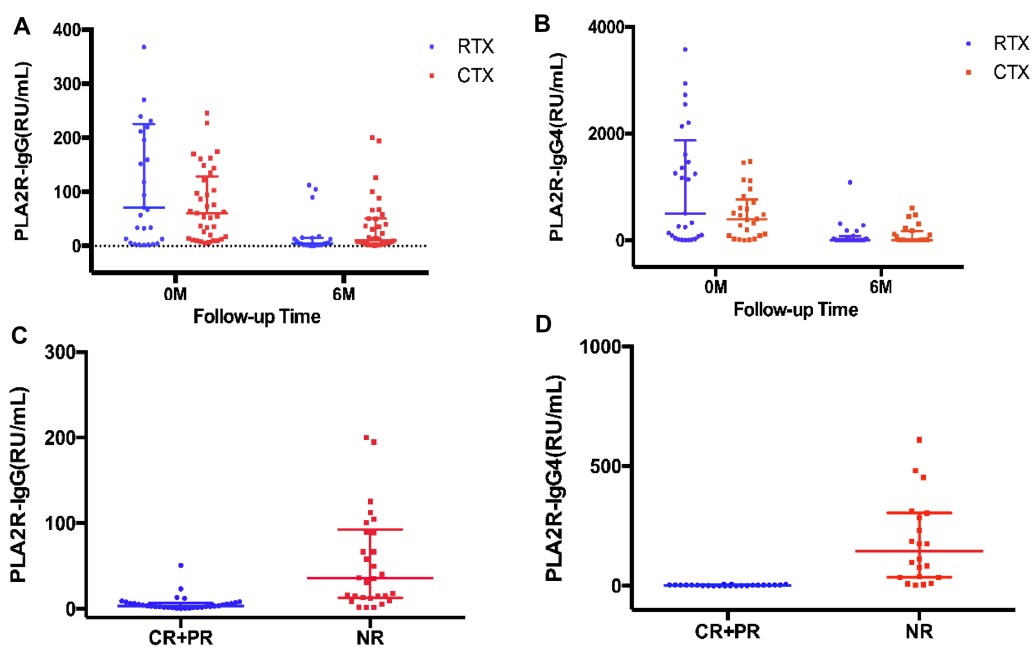

**Figure 2 Effects on PLA2R-IgG and PLA2R-IgG4 levels.** Scatter plots of PLA2R-IgG and PLA2R-IgG4 levels from baseline to the 6th month of the RTX and CTX group (A, B) and of patients achieving complete and partial remission or no remission (C, D) are shown (medians interquartile range).

## Detection of PLA2R-IgG and PLA2R-IgG4 at 6th month for treatment effectiveness analysis and prognostic evaluation in IMN

As shown in Figs. 2C and 2D, the patients who achieved complete or partial remission had a low level of both PLA2R-IgG and PLA2R-IgG4 after six-months of therapy. Further, the ROC curve was analyzed based on the serum PLA2R-IgG and PLA2R-IgG4 levels at 6th month to evaluate their use of them as laboratory criteria for remission (Fig. 3). The AUC of PLA2R-IgG4 and PLA2R-IgG were 0.970 versus 0.886 ($P = 0.0516$), respectively (Fig. 3A). When the cut-off of PLA2R-IgG4 titer was 7.67 RU/mL, the sensitivity and specificity of proteinuria remission at the 6th month were 90.9% and 100%, respectively. Based on the cut-off value of PLA2R-IgG4, 7.67 RU/mL at 6th month, 46 patients were divided into two groups (Table 3A). In the negative group, 24 patients (92.3%) had a remission, while no patient achieved remission in the positive group. Then, we analyzed the value of PLA2R-IgG and PLA2R-IgG4 levels at 6th month using ROC curve to predict the outcome after 12 months of treatment. The AUC of PLA2R-IgG4 and PLA2R-IgG were 0.922 versus 0.897 ($P = 0.3270$), respectively (Fig. 3B). According to the ROC curve, the cut-off value of PLA2R-IgG4 was set at 22.985 RU/mL, and the sensitivity and the specificity of proteinuria remission at the 12th month was 100% and 87.1%, respectively. Forty-four patients were divided into two groups based on this value, as shown in Table 3B, 25 (100%) patients achieved remission in the negative group, and 5 (21.05%) patients achieved remission in the positive group.

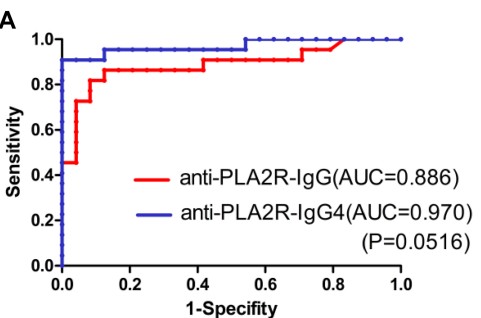
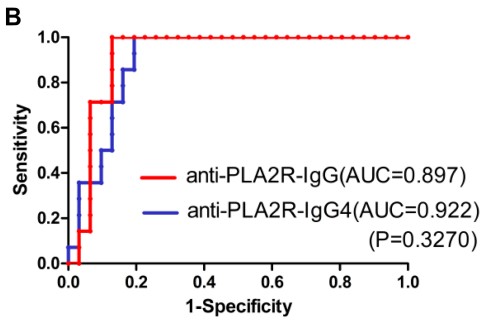

**Figure 3** (A) ROC curve analysis of PLA2R-IgG and PLA2R-IgG4 levels for treatment effectiveness analysis (remission rate at month 6). (B) ROC curve analysis of PLA2R-IgG and PLA2R-IgG4 levels for prognostic evaluation (remission rate at month 12).

**Table 3** (A) The remission rate of proteinuria at 6th month in the positive and negative group. (B) The remission rate of proteinuria at 12th month in the positive and negative groups.

| Outcome | PLA2R-IgG4 (-) | PLA2R-IgG4 (+) |
|---|---|---|
| CR+PR, % | 24/26 (92.3) | 0/20 (0) |
| NR, % | 2/26 (7.7) | 20/20 (100) |

**Notes.**
The cut-off value of PLA2R-IgG4 was 7.67 RU/mL.

| Outcome | PLA2R-IgG4 (−) | PLA2R-IgG4 (+) |
|---|---|---|
| CR+PR, % | 25/25 (100) | 4/19 (21.05) |
| NR, % | 0/25 (0) | 15/19 (78.95) |

**Notes.**
The cut-off value of PLA2R-IgG4 was 22.985 RU/mL.

## Binary loistic regression analysis of remission factors

The sex, age, tubulointerstitial lesion, pathological stages, triglycerides, cholesterol, eGFR, creatinine, albumin, 24-h proteinuria, PLA2R-IgG and PLA2R-IgG4 titers were detected before treatment, and the rate of decline and reduction values of PLA2R-IgG and PLA2R-IgG4 after 6-month therapy were calculated as the relative variables. The decline ratio of PLA2R-igG and PLA2R-IgG4 is the titer at the time of initial treatment minus the titer at the sixth month divided by the titer at the time of initial treatment. Besides, PLA2R-IgG and PLA2R-IgG4 titers were transfored with logarithmic transformation. The Spearman rank-order correlation analysis showed that eGFR, cholesterol, PLA2R-IgG, and PLA2R-IgG4 titers, and the rate of decrease in PLA2R-IgG4 had significant correlations with proteinuria remission at 12th month ($P < 0.05$) (Table 4). Further analysis showed that the titer and rate of decrease of PLA2R-IgG4, as well as eGFR, were better associated with the proteinuria remission than the others. Logistic regression analysis was performed to predict remission rate, which indicated that PLA2R-IgG4 level ($P = 0.023$), the rate of decrease in PLA2R-IgG4 level ($P = 0.034$) and eGFR ($P = 0.012$) were of great importance ($R^2 = 0.716$) (Table 5).

**Table 4  Correlation analyses of remission.**

| Factors | R | *p*-value |
|---|---|---|
| Sex | 0.154 | 0.213 |
| Age | 0.162 | 0.19 |
| Hypertension | 0.079 | 0.524 |
| Tubulointerstitial lesion | 0.122 | 0.326 |
| Pathological stages | 0.034 | 0.787 |
| Serum triglycerides, mmol/L | 0.181 | 0.142 |
| Serum cholesterol, mmol/L | 0.363[**] | 0.003 |
| Uric acid, mmol/L | 0.062 | 0.616 |
| eGFR, mL/min per 1.73 m$^2$ | −0.259[*] | 0.034 |
| Serum creatinine, mmol/L | 0.119 | 0.338 |
| Albumin, g/L | −0.164 | 0.183 |
| 24-h urine protein, g/24 h | 0.09 | 0.468 |
| PLA2R-IgG, RU/mL | 0.266[*] | 0.038 |
| PLA2R-IgG4, RU/mL | 0.300[*] | 0.036 |
| ΔPLA2R-IgG, RU/mL | 0.077 | 0.57 |
| ΔPLA2R-IgG4, RU/mL | 0.137 | 0.37 |
| ΔPLA2R-IgG/PLA2R-IgG | −0.24 | 0.072 |
| ΔPLA2R-IgG4/PLA2R-IgG4 | −0.458[**] | 0.002 |

Notes.
[*]$P < 0.05$.
[**]$P < 0.01$.

**Table 5  Logistic regression analysis.**

| | OR | *p*-value | 95% CI |
|---|---|---|---|
| eGFR | 0.926 | 0.012 | 0.871–0.983 |
| PLA2R-IgG4, RU/mL[a] | 16.720 | 0.023 | 1.486–188.123 |
| ΔPLA2R-IgG4/PLA2R-IgG4 | 0.003 | 0.034 | 0.000–0.646 |

Notes.
[a]PLA2R-IgG4 levels were transformed with logarithmic transformation.

## DISCUSSION

Before the draft of the 2020 new guidelines, CTX has been applied widely as the first-line drug for patients with IMN. According to the 2012 KDIGO guidelines, CTX was recommended for intermediate-risk or high-rick MN patients. The latest guidelines recommended that RTX as the first-line therapy used in intermediate-risk or high-risk MN patients, while CTX should be used preferentially in high-risk or extremely high-risk MN patients. This retrospective study enrolled 70 patients diagnosed with PLA2R-associated IMN from 2016 to 2020 and compared the effectiveness of rituximab and cyclophosphamide combined with a steroid regimen in patients with IMN, who had a moderate or high risk of progressive loss of kidney function and was unlikely to undergo spontaneous remission. No significant differences were found between the two groups after 6 and 12 months of treatment. Specifically, proteinuria remission rates (CR+PR) in the RTX group and CTX group were 44.8% versus 55% ($P = 0.497$) after 6 months of

treatment, and 74% versus 67.5% ($P = 0.114$) after 12 months of treatment, respectively. Specific to CR, the remission rates were 13.8% versus 25% ($P = 0.485$) at the 6th month and 37% versus 52.5% ($P = 0.329$) at the 12th month, respectively, indicating that the regimen of cyclophosphamide combined with steroid may induce CR earlier than rituximab regimen although the results were not statistically different. Similar results were also reported by the RI-CYCLO study (*Scolari et al., 2021*). Regarding the total rate of proteinuria remission after 12 months of treatment, our results were also consistent with some previous randomized controlled trials in which the rates reached 60%–70% (*Fernández-Juárez et al., 2021*; *Fervenza et al., 2019*; *Scolari et al., 2021*). As a retrospective study, the 24-h urine protein in the CTX group was higher than that in the RTX group at the baseline ($4.10 \pm 1.67$ versus $5.71 \pm 2.01$ g/24-h, $P = 0.001$), indicating a higher risk of progressive loss of kidney function. Similar treatment outcomes were achieved with both regimens but CTX may be a better choice for high-risk patients with IMN and help in achieving complete remission earlier. Although the remission had no statistical difference between the two groups, the serum albumin in the RTX group was higher than that in the CTX group at 12th month, demonstrating that the effectiveness of RTX may require long time observation to determine. Given that most studies suggested that IMN patients needed to take a longer time to enter remission, better results would have been emerged in our study by extending the follow-up time (*Fernández-Juárez et al., 2021*; *Fervenza et al., 2019*).

Nowadays the evaluation of the remission is based on 24-h urine protein and serum albumin. But with the discovery of PLA2R and the knowledge advancement on pathogenic mechanism, the PLA2R antibody is regarded as a potential clinical indicator to assess and predict the remission of PLA2R associated IMN, regardless of spontaneous remission or response to immunosuppressive therapy. Although many studies currently are underway or have been published to determine the cut-off value of the PLA2R antibody to predict the remission, no consensus has been reached (*Dahan et al., 2017*; *Jatem-Escalante et al., 2021*). A recent study indicated that when setting a threshold value for PLA2R antibody detected by ELISA at baseline to 97.5 RU/mL had a sensitivity of 71% and a specificity of 81% to predict the spontaneous remission at the 12th month (*Jatem-Escalante et al., 2021*). GEMRITUX trial discovered that anti-PLA2R depletion achieved at 6th month in patients treated with rituximab or supportive treatment was 50% and 12%, respectively. Although there were just 35.1% and 21.1% of patients achieved remission at the 6th month, remission rates were significantly increased at the end of the observational phase (64.9% vs 34.2%) (*Dahan et al., 2017*), indicating that PLA2R antibody levels precede the clinical remission and predict clinical response. Moreover, our previous study found that the efficacy of PLA2R-IgG4 in the diagnosis and risk stratification of IMN was better than that of PLA2R-igG (*Huang et al., 2019*; *Liu et al., 2021*). On the basis of previous work, this study made a preliminary exploration on the clinical efficacy of PLA2R-igG and PLA2R-IgG4 in the prognosis analysis. ROC curve was used to compare the two predictive models of proteinuria remission based on PLA2R-IgG and PLA2R-IgG4 levels, which were detected by TRFIA after 6 months of treatment. The AUC of PLA2R-IgG4 and PLA2R-IgG to predict proteinuria in remission were 0.970 versus 0.886 ($P = 0.0516$) at 6th month, and

0.922 versus 0.897 ($P = 0.3270$) at 12th month, respectively. Further, targeting the cut-off value of PLA2R-IgG4 at 7.67 RU/mL, the sensitivity and specificity of remission at the 6th month were 90.9% and 100%, respectively while with the cut-off value of 22.985 RU/mL, the sensitivity and specificity of remission at 12th month was 100% and 87.1%, respectively. Although there was no statistical difference between the two antibodies, we observed that the AUC of PLA2R-IgG4 in ROC analysis was larger at 6 months and 12 months of treatment, and the sensitivity of PLA2R-IgG4 was higher when the specificity was 100%. The PLA2R-IgG4 had a more accurate trend in predicting remission. In the subsequent work, we will further expand the sample size to find the statistical difference between the PLA2R-igG and PLA2R-IgG4. Logistic regression analysis also indicated that the level and rate of decrease of PLA2R-IgG4 were significantly associated with proteinuria remission. Patients with high PLA2R-IgG titer is associated with bad renal function and not easily remission. *Jatem-Escalante et al. (2021)* found the model including baseline anti-PLA2Rabs and a reduction ≥15% at three months for predicting spontaneous remission with a sensitivity of 93% and a specificity of 80%, which was better than changes of proteinuria in the same period of time. Our study indicated the higher the PLA2R-IgG4 titer and the decline ratio of PLA2R-IgG4 as well as the lower the eGFR, the more likely the inefficacy. The titer and the decline ratio of PLA2R-IgG4 is of great importance with remission.

Our study also had limitations. First, this study was a retrospective study and the baseline characteristics of the two groups could not strictly comply with the characteristics of a randomized controlled trial, which would have limited the significance of the statistical analysis of therapeutic evaluation. Second, the endpoint of our study was the remission rate of proteinuria after immunosuppressive therapy for 12 months, and we may have had a better conclusion of remission and may have observed the incidence of relapse if follow-up time would have extended because RTX needs a longer time to respond according to previous studies (*Fernández-Juárez et al., 2021*; *Fervenza et al., 2019*). Third, our study was a single-center study, the results were based on small population, and the findings may not have been highly reliable. The follow-up time should be prolonged and the number of the patients should be increased in future studies.

## CONCLUSIONS

In conclusion, the patients in both the RTX group and CTX group achieved effective remission of proteinuria after 12 months of treatment. Compared with PLA2R-IgG, PLA2R-IgG4 may be a better effective biomarker for treatment effectiveness analysis and prognostic assessment in PLA2R associated IMN.

### Funding

This study was funded by the Wuxi Medical Innovation Team Project (CXTD2021010), the Jiangsu Province "333" project (BRA2020142), the National Natural Science Foundation of China (Youth Program 81900698), the Natural Science Foundation of Jiangsu Province

(Grant NO. BK20210067), the Top Talent Support Program for young and middle-aged people of Wuxi Health Committee (HB2020008), the Major projects of precision medicine of Wuxi Health Committee (J202001), the Chinese National Natural Science Foundation (No. 82172336), and the Key Research and Development Project of Zhejiang Province (No. 2022C03118). The funders had no role in study design, data collection and analysis, decision to publish, or preparation of the manuscript.

### Grant Disclosures

The following grant information was disclosed by the authors:
Wuxi Medical Innovation Team Project: CXTD2021010.
Jiangsu Province ''333'' project: BRA2020142.
National Natural Science Foundation of China: 81900698.
Natural Science Foundation of Jiangsu Province: BK20210067.
Top Talent Support Program: HB2020008, J202001.
Chinese National Natural Science Foundation: 82172336.
Key Research and Development Project of Zhejiang Province: 2022C03118.

### Competing Interests

The authors declare there are no competing interests.

### Author Contributions

- Yiqing Huang performed the experiments, analyzed the data, prepared figures and/or tables, authored or reviewed drafts of the article, and approved the final draft.
- Junyi Zhou performed the experiments, prepared figures and/or tables, authored or reviewed drafts of the article, and approved the final draft.
- Kezhi Zhou performed the experiments, prepared figures and/or tables, authored or reviewed drafts of the article, and approved the final draft.
- Biao Huang analyzed the data, authored or reviewed drafts of the article, and approved the final draft.
- Jing Xue analyzed the data, authored or reviewed drafts of the article, and approved the final draft.
- Xiran Zhang analyzed the data, authored or reviewed drafts of the article, and approved the final draft.
- Bin Liu conceived and designed the experiments, authored or reviewed drafts of the article, and approved the final draft.
- Zhijian Zhang performed the experiments, authored or reviewed drafts of the article, and approved the final draft.
- Leting Zhou analyzed the data, authored or reviewed drafts of the article, and approved the final draft.
- Ting Cai performed the experiments, authored or reviewed drafts of the article, and approved the final draft.
- Yi Zhang analyzed the data, authored or reviewed drafts of the article, and approved the final draft.

- Zhigang Hu analyzed the data, authored or reviewed drafts of the article, and approved the final draft.
- Liang Wang conceived and designed the experiments, authored or reviewed drafts of the article, and approved the final draft.
- Xiaobin Liu conceived and designed the experiments, analyzed the data, prepared figures and/or tables, authored or reviewed drafts of the article, and approved the final draft.

## Human Ethics

The following information was supplied relating to ethical approvals (i.e., approving body and any reference numbers):

This study was approved by the ethics committee of the Wuxi People's Hospital (approval no. kyl2016001).

## Data Availability

The raw data is available in the Supplementary Files.

## Supplemental Information

Supplemental information for this article can be found online at http://dx.doi.org/10.7717/peerj.14193#supplemental-information.

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
