# Peer review of "PLA2R-IgG4 antibody as a predictive biomarker of treatment effectiveness and prognostic evaluation in patients with idiopathic membranous nephropathy: a retrospective study"

_PeerJ, doi:10.7717/peerj.14193_

## Round 0.1 · original submission · Major Revisions

There is a split decision among the reviewers here. reviewer-3 is firmly negative in their assessment, and raises a number of crucial issues. I am willing to give you an opportunity to address these, but you will need to satisfy this reviewer in your response.

I note that PeerJ does not discriminate on the basis of 'novelty', and replicative studies are deemed appropriate content. For this reason, I am happy for you to disregard point #4 of this critique. However, ALL other issues must be carefully addressed.

The work will need to go back to reviewer-3 for a further opinion so I would urge you to address these points carefully.

Reviewer 1 ·

Basic reporting

No comment.

Experimental design

No comment.

Validity of the findings

No comment.

Additional comments

The study presented the regimen of Rituximab and prednisone combined with cyclophosphamide have the same efficacy in the remission of proteinuria after 12 months of treatment in IMN. Besides, PLA2R has been a hot topic in the past few years, and this new work suggests PLA2R-IgG4 may be a better biomarker for treatment effectiveness and prognostic evaluation in IMN creatively. It is a meaningful and interesting topic and may have certain clinical significance in treatment and prognosis of IMN. However there are still some shortcomings and issues,as detailed in the follows.
1.The correlation analysis showed the titer and rate of decrease of PLA2R-IgG4 were significantly associated with proteinuria remission. The authors could discuss more detail about the remission factors to make it clear and concise for readers in the discussion.
2.As is known to all, the most used method for the detection of PLA2R antibody is ELISA. This study applied a new method named TRFIA, and detected the PLA2R-IgG4 titer. Whether it has been a new reliable test method used in the clinical utility?
3.There are some spelling and grammar mistakes in the manuscript, which need to be checked carefully, such as line 38 “Receiver operating characteristic curve (ROC) curve...”.The Legend of figure 2 is lengthy and jumbled, which should be simplified. The title of table 3 has a definite mistake,which should be corrected.

·

Basic reporting

I commend the authors for their extensive data set, compiled over many years of detailed fieldwork. In addition, the tables and figures of the manuscript are concise and explicit. But the English language needs to be reviewed, by a fluent speaker if possible.

Experimental design

The authors mentioned a meaningful point of view that PLA2R-IgG4, compared with PLA2R-IgG, may be a better biomarker for the treatment effectiveness analysis and prognostic assessment. I have the following questions before Acceptance.
1. Why the authors chose the method of “TRFIA” to detect antibodies of PLA2R rather than ELISA which has been routinely used in clinical practice after all. It would be better if the authors could provide more details about the selection of “TRFIA”.
2. In this study, patients with IMN treated with either “CTX” or “RTX”. Why the other treatments, such as FK506/CsA, are not applied?

Validity of the findings

no comment

Additional comments

There are some clerical errors.
1.In row 107,the storage temperature might be changed to -80℃.
2.In table 1,serum creatinine may be calculated in μmol/L.

Reviewer 3 ·

Basic reporting

This study raise an interesting biomarker PLA2R-IgG4 antibody as a predictive indicator of treatment effectiveness and prognostic evaluation in patients with idiopathic membranous nephropathy, but it is not qualified to be published in the journal in my opinion.

Experimental design

The purpose of the study was not clear enough. It seems that the authors present two main objectives. One is to compare the remission rate of proteinuria between the RTX group and CTX group after 12 months of treatment. Another is to clarify that PLA2R-IgG4 had better effectiveness and prognostic value over PLA2R-IgG.However, there were several issues. First, in accordance with the KDIGO guidelines, the type of treatment must be adapted to the the risk of progressive loss of kidney function of each patient, based in clinical and laboratory data. Enrolled patients in the current study seemed had moderate risk based on baseline proteinuria and eGFR level,which should be given RTX or CNI±glucocorticoids.CTX regimen should be restricted to patients considered at high or very high risk. Second, insufficiency pathological features were given. Deposition of PLA2R,IgG1-4, etc. should be supplemented.Third, What is the method to calculate the rate of decrease in PLA2R-IgG4 level? It should be clarified in the Methods. It may be more appropriate to determin risk factors of CR by univariate and multivariate Cox regression.

Validity of the findings

# Line 260-263 the authors delcare that “The AUC of PLA2R-IgG and PLA2R-IgG4 to predict proteinuria in remission were 0.970 versus 0.886 (P=0.0516) at 6th month, and 0.922 versus 0.897 (P=0.3270) at 12th month, respectively”. The AUC of PLA2R-IgG was larger than PLA2R-IgG4?The description was inconsistent with the previous results.

# What is the advantage of PLA2R-IgG4 detection for clinical use? Is it more convenient or inexpensive to test since no significant statistical differences were found between PLA2R-IgG and PLA2R-IgG4 levels to predict proteinuria remission by ROC curve.

# The conclusions of the study had no much difference from previous studies.What new information can readers get from the current study?

Additional comments

This study raise an interesting biomarker PLA2R-IgG4 antibody as a predictive indicator of treatment effectiveness and prognostic evaluation in patients with idiopathic membranous nephropathy, but it is not qualified to be published in the journal in my opinion.

1#The purpose of the study was not clear enough. It seems that the authors present two main objectives. One is to compare the remission rate of proteinuria between the RTX group and CTX group after 12 months of treatment. Another is to clarify that PLA2R-IgG4 had better effectiveness and prognostic value over PLA2R-IgG.However, there were several issues. First, in accordance with the KDIGO guidelines, the type of treatment must be adapted to the the risk of progressive loss of kidney function of each patient, based in clinical and laboratory data. Enrolled patients in the current study seemed had moderate risk based on baseline proteinuria and eGFR level,which should be given RTX or CNI±glucocorticoids.CTX regimen should be restricted to patients considered at high or very high risk. Second, insufficiency pathological features were given. Deposition of PLA2R,IgG1-4, etc. should be supplemented.Third, What is the method to calculate the rate of decrease in PLA2R-IgG4 level? It should be clarified in the Methods. It may be more appropriate to determin risk factors of CR by univariate and multivariate Cox regression.

2# Line 260-263 the authors delcare that “The AUC of PLA2R-IgG and PLA2R-IgG4 to predict proteinuria in remission were 0.970 versus 0.886 (P=0.0516) at 6th month, and 0.922 versus 0.897 (P=0.3270) at 12th month, respectively”. The AUC of PLA2R-IgG was larger than PLA2R-IgG4?The description was inconsistent with the previous results.

3# What is the advantage of PLA2R-IgG4 detection for clinical use? Is it more convenient or inexpensive to test since no significant statistical differences were found between PLA2R-IgG and PLA2R-IgG4 levels to predict proteinuria remission by ROC curve.

4# The conclusions of the study had no much difference from previous studies.What new information can readers get from the current study?

---

## Round 0.2 · Major Revisions

While I realise you have addressed many of the comments in the rebuttal, these have not resulted in sufficient explanation in your revised paper, so I am returning this to you for further amendment. Please include clear statements and/or explanations on the points I include below in your manuscript.




Reviewer1
1.Q: The correlation analysis showed the titer and rate of decrease of PLA2R-IgG4 were significantly associated with proteinuria remission. The authors could discuss more detail about the remission factors to make it clear and concise for readers in the discussion.
A: Thank you for comment which is quite helpful for this manuscript. Conclusion in this study, the higher the PLA2R-IgG4 titer and decrease rate of PLA2R-IgG4 as well as the lower the eGFR, the more likely the inefficacy remission. And it is added in the discussion at line 278-279.

This needs more depth and clear discussion.



Reviewer2
1.Q: Why the authors chose the method of “TRFIA” to detect antibodies of PLA2R rather than ELISA which has been routinely used in clinical practice after all. It would be better if the authors could provide more details about the selection of “TRFIA”.
A: Taking advantage of the long decay ….. we have determined TRFIA had higher sensitivity than ELISA which improving the sensitivity to 74% and specificity was 100% in diagnosing IMN and further improving the sensitivity to 90% by investigating PLA2R-IgG4 titer. The sensitivity was far more than which detected by ELISA.

This information should be clearly described in the paper and cited, and if it has not been published elsewhere should be included as supplementary material.


Reviewer3

Q: In accordance with the KDIGO guidelines, the type of treatment must be adapted to the risk of progressive loss of kidney function of each patient, based in clinical and laboratory data. Enrolled patients in the current study seemed had moderate risk based on baseline proteinuria and eGFR level,which should be given RTX or CNI±glucocorticoids. CTX regimen should be restricted to patients considered at high or very high risk.
A: This study was a retrospective study, and the enrolled patients were hospitalized from 2016 to 2020. Before the draft of the 2020 new guidelines, the diagnosis and treatment were according to the 2012 KDIGO guidelines. Immunosuppressive therapy is recommended for intermediate-risk patients who have not responded for 6 months expectant treatment or high-risk patients. In the selection of immunosuppressive agents, CTX is the first choice. Since the update of the new draft guideline of KDIGO in 2020, we have updated the diagnosis and treatment concept according to the guidance of the new guideline: RTX should be used in intermediate-risk or high-risk MN patients, while CTX should be used preferentially in high-risk or extremely high-risk MN patients. Thanks for your pertinent suggestion.

This must be clearly explained in the manuscript. Your argument is fine, but it must be built into the manuscript text.


Q: Insufficiency pathological features were given. Deposition of PLA2R,IgG1-4, etc. should be supplemented.
A: In the study, the patients were PLA2R-associated membranous nephropathy confirmed by renal biopsy. The immunofluorescence was positive for PLA2R and mainly IgG4 subclass deposition. The related content has been supplemented in the article at line 100-101. Thanks again for your advice.

Insufficient depth and detail provided. Please add detail.




Q: What is the advantage of PLA2R-IgG4 detection for clinical use? Is it more convenient or inexpensive to test since no significant statistical differences were found between PLA2R-IgG and PLA2R-IgG4 levels to predict proteinuria remission by ROC curve.

A: In 2009, Beck found that PLA2R and IgG4 are co-deposited in glomeruli. Recent studies have found that the pathogenesis of MN is caused by PLA2R-IgG4 activating complement through lectin pathway, leading to podocyte injury and proteinuria [1]. The risk stratification model of MN in the 2021 edition of KDIGO guideline only lists some biomarkers with evidence-based evidence. The clinical role of PLA2R-IgG4 in the diagnosis, risk stratification and prognosis analysis of MN is still inconclusive. The precise diagnosis and treatment of MN still needs to be explored. Our previous study found that the efficacy of PLA2R-IgG4 in the diagnosis and risk stratification of IMN was better than that of PLA2R-igG [2-4]. On the basis of previous work, this study made a preliminary exploration on the clinical efficacy of PLA2R-igG and PLA2R-IgG4 in the prognosis analysis, although there was no statistical difference between the two antibodies. However, we observed that the AUC of PLA2R-IgG4 in ROC analysis was larger at 6 months and 12 months of treatment, and the sensitivity of PLA2R-IgG4 was higher when the specificity was 100%. The PLA2R-IgG4 had a more accurate trend in predicting remission. In the subsequent work, we will further expand the sample size to find the statistical difference between the PLA2R-igG and PLA2R-IgG4.

Please include this in the relevant section of your paper.

---

## Round 0.3 · accepted · Accept

Thanks for attending to these remaining issues.